# DOMAIN GENERALIZATION VIA PARETO OPTIMAL GRADIENT MATCHING

## ABSTRACT

In this study, we address the gradient-based domain generalization problem, where predictors aim for consistent gradient directions across different domains. Existing methods have two main challenges. First, minimization of gradient empirical distance or gradient inner products (GIP) leads to *gradient fluctuations* and *magnitude elimination* among domains, thereby hindering straightforward learning. Second, the direct application of gradient learning to joint loss function can incur *high computation overheads* due to second-order derivative approximation. To tackle these challenges, we propose a new Pareto Optimality Gradient Matching (POGM) method. In contrast to existing methods that add gradient matching as regularization, we leverage gradient trajectories as collected data and apply independent training at the meta-learner. In the meta-update, we maximize GIP while limiting the learned gradient from deviating too far from the empirical risk minimization gradient trajectory. By doing so, the aggregate gradient can incorporate knowledge from all domains without suffering gradient magnitude elimination or fluctuation towards any particular domain. Experimental evaluations on datasets from DomainBed demonstrate competitive results yielded by POGM against other baselines while achieving computational efficiency.

## 1 INTRODUCTION

Domain generalization (DG) has emerged as a significant research field in machine learning, owing to its practical relevance and parallels with human learning in new environments. In DG frameworks, learning occurs across multiple datasets collected from diverse environments, with no access to data from the target domain (Zhou et al., 2023). Various strategies have been proposed to address DG challenges, including distributional robustness (Eastwood et al., 2022a), domain-invariant representations (Li et al., 2018; Zhao et al., 2020; Bui et al., 2021), invariant risk minimization (Nguyen et al., 2021; 2022), and data augmentation (Yao et al., 2022; Zhou et al., 2021; Li et al., 2021).

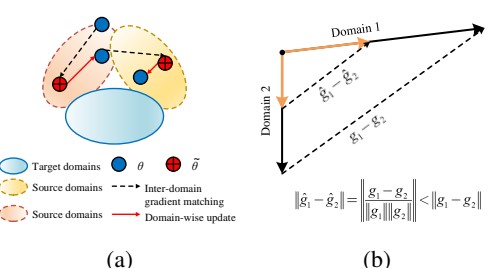

Figure 1: Issues in current gradient-based DG methods. (a) We observe that Fish (Shi et al., 2022) induces *gradient fluctuations* across domains, hindering straightforward convergence to the global optimum due to its simplistic optimization approach. (b) We show that the optimal solution for Fishr (Rame et al., 2022) is obtained when *gradient magnitudes are eliminated* in each domain, minimizing empirical gradient distance.

Recently, gradient-based DG has emerged as a promising approach to enhance generalization across domains. This approach is *orthogonal* to the previously mentioned methods in nature, allowing for their *combined integration to facilitate additional performance improvements*. Particularly, gradient-based DG's target is discovering an invariant gradient direction across source domains. Fish (Shi et al., 2022) introduces a gradient inner product (GIP) to ensure consistent gradient trajectories among domains, whereas Fishr (Rame et al., 2022) minimizes the Euclidean distance between domains'

gradient trajectories. These methods increase the likelihood that the gradient direction remains consistent even on unseen target domains. In this paper, we perform a theoretical and empirical analysis to show that there exists the potential degradation in the learning progress of these methods due to *gradient fluctuation* and *gradient magnitude elimination* (see the concept in Fig. 1). We show that gradient magnitude elimination is caused by using MSE as regularization among gradient pairs as proposed in Fishr while gradient fluctuation is caused by updating each domain sequentially in Reptile (Nichol et al., 2018) based Fish (Shi et al., 2022, Alg. 1). Additionally, employing Hessian approximations, as in Fish (Shi et al., 2022, Alg. 2), incurs substantial computational overhead.

To address these challenges, we introduce Pareto Optimality Gradient Matching (POGM), a simple but effective gradient-based domain generalization method. **First**, to mitigate gradient magnitude elimination and prevent gradient fluctuation phenomenon, we formulate our gradient matching by summing over pairs of gradient inner product (GIP), and we confine the search space for the GIP solution to a $\kappa$-hypersphere centered around the ERM gradient trajectory to reduce the effort in finding optimal solutions. **Second**, to solve the summation of GIP of gradients over $K$ domains, which has a complexity of $\mathcal{O}(K \times (K-1)/2)$, we first utilize the Pareto front to transform the task of minimizing all GIP pairs into focusing solely on the worst-case scenario. Then, we introduce a closed-form relaxation method for inter-domain gradient matching. As a result, the complexity of our POGM can be reduced to $\mathcal{O}(2 \times K)$. **Third**, to circumvent the computational overhead associated with Hessian approximations, we leverage meta-learning and consider gradient matching as a separate process. Hence, our method can learn a set of coefficients for combining domain-specific gradients with scaled weights. As a consequence, POGM can approximate weighted aggregated domain-specific gradient updates without the need for second-order derivatives. Our experiments show that POGM achieves state-of-the-art performance across datasets from the recent DG benchmark, DomainBed (Gulrajani & Lopez-Paz, 2021). The robust performance of our method across diverse datasets underscores its broad applicability to different applications and sub-genres of DG tasks.

## 2 RELATED WORKS

Several approaches for DG have been explored, which can be broadly classified into two categories: finding domain-invariant representation and representation mixing. From the perspective of the first category, Mahajan et al. (2021) introduces a regularization-based framework to generate invariant representations by minimizing the empirical distance between encoded representations from different domains. Nguyen et al. (2022; 2021) minimize the empirical distance between the source and target data distribution. Nguyen et al. (2021) assumed that the target dataset is not accessible, and therefore proposed to generate data on the target dataset using a generative adversarial network (GAN). However, this approach is applicable within limited constraints, where the target dataset is required to be accessible to the GAN model. Li et al. (2018); Zhao et al. (2020); Bui et al. (2021) propose GAN-based frameworks for learning invariant representations. These works highlight the redundancy in classifier networks, leading to the introduction of a discriminator network to enhance the extraction of meaningful information. Additionally, Kim et al. (2021) suggests leveraging self-supervised learning to generate domain-invariant representations. Lv et al. (2022) applies Barlow Twins to generate causal invariant representations, assumed to be domain-invariant. According to the second category, Shu et al. (2021); Yao et al. (2022); Zhou et al. (2021); Yan et al. (2020) leverages mixing strategy to inter-domain representations to improve the DG. Li et al. (2021) proposes a simple data augmentation approach by perturbing the latent features with the white Gaussian noise.

SWAD (Cha et al., 2021) propose a different approach for DG by leveraging stochastic weight averaging to smoothen the loss landscape, thus, improving the generalization.

Recently, Shi et al. (2022) pioneers gradient-based DG, introducing Fish to discover invariant gradient trajectories across domains for enhanced model consistency amidst domain shifts, thereby improving generalization to unseen datasets. Additionally, Rame et al. (2022) introduces Fishr to enhance gradient learning by incorporating gradient variant regularization into the loss function, capturing both the first and second moments of the gradient distribution, thereby leveraging richer information for gradient learning. Our work is also a gradient-based DG method, in which we aim to address the gradient limitations in Fish and Fishr by exploiting Pareto optimality for gradient matching.

# 3 A THEORETICAL AND EMPIRICAL ANALYSIS ON FISH AND FISHR

## 3.1 PROBLEM SETTINGS

Let $\mathcal{X}$ and $\mathcal{Y}$ be the feature and label spaces, respectively. There are $K$ source domains $\mathcal{K} = \{D_i\}_{i=1}^K$ and $L$ target domains $\{D_i\}_{i=K+1}^{K+L}$. The goal is to generalize the model learned using data samples of the source domains to unseen target domains. Herein, we denote the joint distribution of domain $i$ by $P_i(X, Y)$ $(X, Y \sim \mathcal{X}, \mathcal{Y})$. During training, there are $K$ datasets $\{S_i\}_{i=1}^K$ available, where $S_i = \{(x_j^{(i)}, y_j^{(i)})\}_{j=1}^{N_i}$, $N_i$ is the number of samples of $S_i$ that are sampled from the $i^{\text{th}}$ domain. At test time, we evaluate the generalization capabilities of the learned model on $L$ datasets sampled from the $L$ target domains, respectively. This work focuses on DG for image classification where the label space $\mathcal{Y}$ contains $C$ discrete labels $\{1, 2, \ldots, C\}$.

To facilitate our analysis, we model a gradient-based domain generalization (GBDG) algorithm that aims to learn an invariant gradient trajectory via the generalized loss $\mathcal{L}_{\text{GBDG}} = \mathcal{L}_{\text{ERM}} + \lambda \mathcal{L}_{\text{IG}}$, where $\mathcal{L}_{\text{GBDG}}$ is denoted as GBDG loss, $\mathcal{L}_{\text{ERM}}$ is the empirical risk minimization (ERM) (Vapnik, 1998) loss function. $\mathcal{L}_{\text{IG}}$ is the invariant gradient regularizer, which can represent both Fish (Shi et al., 2022) or Fishr (Rame et al., 2022), and can be defined as $\mathcal{L}_{\text{IG}} = \sum_{i,j \in \mathcal{K}} \mathcal{L}_{\text{dist}}(\nabla \mathcal{L}_i(\theta), \nabla \mathcal{L}_j(\theta))$, where $\mathcal{L}_{\text{dist}}$ denotes the empirical distance between two gradient vectors of the model parameters $\theta$ trained from domains $i$ and $j$, respectively. We denote $\mathcal{L}_i$ as the loss according to the training on a domain $i \in \mathcal{K}$. $\mathcal{L}_{\text{dist}}$ is computed via GIP and by mean-square error (MSE) in Fish and Fishr, respectively. To understand Fish and Fishr, we conduct both theoretical analysis and empirical experiments on three baselines including Fish, Fishr, and the conventional ERM. The detailed setup of the experiments is demonstrated in Appendix. C.1.

## 3.2 GRADIENT MAGNITUDE ELIMINATION

Let us begin with an analysis of Fishr (Rame et al., 2022) by identifying the issue of gradient magnitude elimination via Lemma 1.

**Lemma 1 (Gradient Magnitude Elimination)** *One of the feasible solutions of* $\min_\theta \sum_{i,j \in \mathcal{K}} \mathcal{L}_{dist}\left(\nabla \mathcal{L}_i(\theta), \nabla \mathcal{L}_j(\theta)\right)$ *in Fishr is when* $\|\nabla \mathcal{L}_i(\theta)\|, \|\nabla \mathcal{L}_j(\theta)\| \sim 0$.

Lemma 1 reveals that the optimization problem from (Rame et al., 2022, Eq. 4) can lead to the worst case where the $\|\nabla \mathcal{L}_i(\theta)\|, \|\nabla \mathcal{L}_j(\theta)\| \sim 0$. In other words, Fishr does not reduce the angles among the Fishr gradient and the domain-wise gradients (Figs. 2a, 2b). Meanwhile, the norm differences between the Fishr model and the domain-specific models are reduced (in contrast with the angle results). This means that the gradient norm on each domain-specific gradient is reduced instead of reducing the angle among the gradient as claimed in (Shi et al., 2022; Rame et al., 2022) although the penalty is reduced (Fig. 2c). As a result, the task loss of Fishr is worse than the ERM (Fig. 2d). Detailed analysis is demonstrated in Appendix C.3.

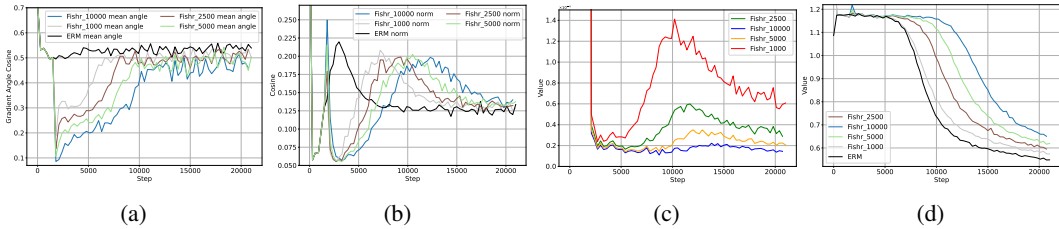

| (a) | (b) | (c) | (d) |

Figure 2: The empirical analysis on Fishr according to the angle between the learned invariant gradient and the domain-wise gradients (Fig. 2a), and norm difference between Fishr-trained and domain-specific models (Fig. 2b). The comparison of training loss and penalty value of Fishr are demonstrated in Figs. 2c, 2d, respectively.

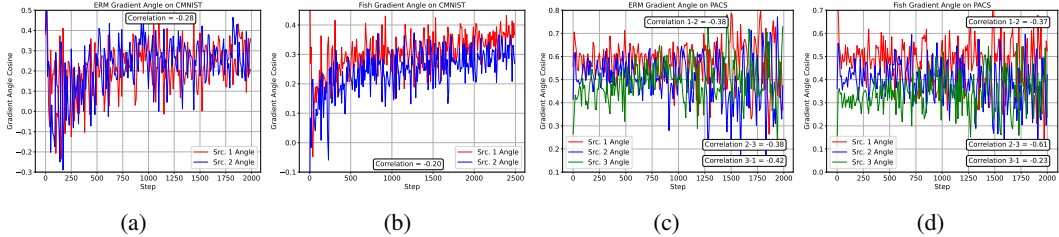

Figure 3: The gradient direction angles of ERM and Fish in CMNIST, PACS. Although it is claimed that the Fish gradient is invariant to domain-specific gradients, the angles between the Fish gradient and domain-specific gradients tend to fluctuate considerably. This leads to a lower correlation among domain-specific gradients compared to the ERM gradient.

### 3.3 GRADIENT FLUCTUATION

We show that although the conventional Fish (Shi et al., 2022, Alg. 2) aims to reduce the gradient angles via GIP, the Reptile-based Fish (Shi et al., 2022, Alg. 1) yields significantly fluctuated gradient directions (i.e., lower correlation among angles' cosines). This can be observed in both the CMNIST and PACS datasets (see Fig. 3), where Fish shows no significant improvement compared with the conventional ERM.

## 4 POGM: GRADIENT MATCHING VIA PARETO OPTIMALITY

### 4.1 GRADIENT INNER PRODUCT WITH GENERALIZED CONSTRAINTS

Our method aims to simultaneously alleviate gradient magnitude elimination (Section 3.2) and address the issues of gradient fluctuation (Section 3.3). We adopt GIP as proposed from Fish (Shi et al., 2022) but aim to restrict the searching space for our GIP problem within a $\kappa$-hypersphere, which has the center determined by the ERM trajectory. Specifically, we propose the GIP with generalized constraints (GIP-C) as follows:

$$\mathcal{L}_{\text{GIP-C}} = \sum_{i \in \mathcal{K}} \sum_{\substack{j \in \mathcal{K} \\ i \neq j}} \nabla \mathcal{L}_i(\theta) \cdot \nabla \mathcal{L}_j(\theta) - \gamma \Big( \|\nabla \mathcal{L}_{\text{GIP-C}} - \nabla \mathcal{L}_{\text{ERM}}\|^2 - \kappa \|\nabla \mathcal{L}_{\text{ERM}}\|^2 \Big), \quad (1)$$

where $\mathcal{L}_{\text{ERM}}$ is the ERM loss defined as $\mathcal{L}_{\text{ERM}} = \frac{1}{K} \sum_{i=1}^{K} \mathcal{L}_i(\theta)$. We utilize ERM as a standard gradient trajectory as ERM is simple, straightforward, and demonstrates good results in DG. The advantage of adding this constraint is twofold: **1)** The learned invariant gradient $\nabla \mathcal{L}_{\text{GIP-C}}$ is not biased to one set of domains, which ensures the generalization of our algorithm. **2)** The learned invariant gradient $\nabla \mathcal{L}_{\text{GIP-C}}$ is either backtracked towards the previous checkpoint or biased towards specific domains with a value larger than $\kappa \|\nabla \mathcal{L}_{\text{ERM}}\|$.

However, the incorporation of loss as depicted in Eq. (1) necessitates a second-order derivative approximation and becomes NP-hard. Additionally, the algorithm has a complexity of $\mathcal{O}(K \times (K - 1)/2)$ as it sums over all pairs of gradients in $K$ domains. Consequently, the learning is hampered by both the computational overhead and the challenge of identifying the optimal solution.

In the next section, our target advances twofolds. First, to alleviate the NP-hard problem, we propose a relaxation for the optimization problem to reduce the computation complexity. Second, to mitigate computational overhead caused by approximating the second-order derivative, we utilize meta-learning (Finn et al., 2017) to establish a unified learning framework for our DG process.

### 4.2 RELAXATION OF INTER-DOMAIN GRADIENT MATCHING

By finding $\theta_{\text{GIP-C}}$ using Eq. (1), we can find $\theta_{\text{GIP-C}}$ that achieve the angles between gradients induced by $\theta_{\text{GIP-C}}$ to the gradients on source domains $D_k, \forall i \in \mathcal{K}$ are maximized. For instance,

$$\theta_{\text{GIP-C}} = \arg \max_{\theta} \sum_{i \in \mathcal{K}} \nabla \mathcal{L}_i(\theta) \cdot \nabla \mathcal{L}_{\text{GIP-C}} - \gamma \Big( \|\nabla \mathcal{L}_{\text{GIP-C}} - \nabla \mathcal{L}_{\text{ERM}}\|^2 - \kappa \|\nabla \mathcal{L}_{\text{ERM}}\|^2 \Big). \quad (2)$$

Herein, Eq. (2) is a multi-objective optimization problem. In general, no single solution can optimize all objectives at the same time. To overcome this, we determine the Pareto front that provides a trade-off among the different objectives. We consider the following definitions (Zitzler & Thiele, 1999):

**Definition 1 (Pareto dominance)** *Let $\theta^a, \theta^b \in R^m$ be two points, $\theta^a$ is said to be dominated $\theta^b$ ($\theta^a \succ \theta^b$) if and only if $\mathcal{L}_i(\theta^a) \leq \mathcal{L}_i(\theta^b), \forall i \in \mathcal{K}$ and $\mathcal{L}_j(\theta^a) < \mathcal{L}_j(\theta^b), \exists j \in \mathcal{K}$.*

**Definition 2 (Pareto optimality)** *$\theta^*$ is a Pareto optimal point and $\mathcal{L}(\theta^*)$ is a Pareto optimal objective vector if it does not exist $\hat{\theta} \in R^m$ such that $\hat{\theta} \prec \theta^*$. The set of all Pareto optimal points is called the Pareto set. The image of the Pareto set in the loss space is called the Pareto front.*

To leverage Definition 2, we present the following lemma:

**Lemma 2** *The average cosine similarity between the given gradient vector $\nabla\mathcal{L}_{GIP\text{-}C}$ and the domain-specific gradient is lower-bounded by the worst-case cosine similarity as follows:*

$$\frac{1}{K} \sum_{i \in \mathcal{K}} \nabla\mathcal{L}_i(\theta) \cdot \nabla\mathcal{L}_{GIP\text{-}C} \geq \min_{i \in \mathcal{K}} \nabla\mathcal{L}_i(\theta) \cdot \nabla\mathcal{L}_{GIP\text{-}C}.$$

Lemma 2 allows the realization that the maximization of our multi-objective function can be reduced to maximizing the worst-case scenario. The approach leads us to attain the optimal Pareto front. Hence, the following lemma follows:

**Lemma 3** *Given $\theta^*$ as an Pareto optimal solution of $\theta$, we have $\theta^*$, which is also the solution of*

$$\max_{\theta} \min_{i \in \mathcal{K}} \left[ \nabla\mathcal{L}_i(\theta) \cdot \nabla\mathcal{L}_{GIP\text{-}C}(\theta) - \gamma\Big( \|\nabla\mathcal{L}_{GIP\text{-}C}(\theta) - \nabla\mathcal{L}_{ERM}(\theta)\|^2 - \kappa\|\nabla\mathcal{L}_{ERM}(\theta)\|^2 \Big) \right] \quad (3)$$

Eq. (3) represents the optimization over the gradient. To simplify implementation and reduce computational complexity, we seek optimal gradients along gradient trajectories. The approach enables us to circumvent noise introduced by mini-batch gradients, ensuring both optimization accuracy and stability. Specifically, we define the gradient trajectory of domain $i$ and define the ERM gradient trajectory as

$$h_i^{(r)} = \theta_i^{(r+1)} - \theta^{(r)} = \sum_{e=1}^{E} \nabla\mathcal{L}_i(\theta^{(r,e)}), \;\; h_{ERM}^{(r)} = \theta_{ERM}^{(r+1)} - \theta_{ERM}^{(r)} = \frac{1}{K} \sum_{i=1}^{K} \sum_{e=1}^{E} \nabla\mathcal{L}_i(\theta^{(r,e)}). \quad (4)$$

From Lemma 3 on Pareto optimal solution, we derive the theorem for an invariant gradient solution as follows:

**Theorem 1 (Invariant Gradient Solution)** *Given the Pareto condition as mentioned in Lemma 3, $\pi = \{\pi_1^{(r)}, \dots, \pi_K^{(r)}\}$ are the set of K learnable scaling parameters, which coordinate the domain-wise gradient trajectory $h_i^{(r)}, \forall i \in \mathcal{K}$ at each training iteration. The invariant gradient $h_{GIP\text{-}C}$ is characterized by:*

$$h_{GIP\text{-}C}^{(r)} = h_{ERM}^{(r)} + \frac{\kappa\|h_{ERM}^{(r)}\|}{\|h_{\widetilde{\pi}}^{(r)}\|} h_{\widetilde{\pi}}^{(r)} \quad s.t. \quad \widetilde{\pi} = \arg\min_{\pi} h_{\pi}^{(r)} \cdot h_{ERM}^{(r)} + \sqrt{\kappa}\|h_{ERM}^{(r)}\|\|h_{\pi}^{(r)}\|, \quad (5)$$

*where $h_{\pi}^{(r)} = \sum_{i=1}^{K} \pi_i^{(r)} h_i^{(r)}, \; \sum_{i=1}^{K} \pi_i^{(r)} = 1$. We denote $\widetilde{\pi}$ as the optimal parameter set at round $r$.*

**Remark 1** *The computation of the loss function using Theorem 1 reduces to $\mathcal{O}(2 \times K)$ as we only need to compute the GIP between two aggregated gradients once.*

From Theorem 1, GIP-C appears to have a close relationship with ERM. For instance,

**Corollary 1** *When the radius of the $\kappa$-hypersphere reduces to $0$, the GIP-C is reduced to ERM. For instance, $\lim_{\kappa \to 0} h_{GIP\text{-}C} = h_{ERM}^{(r)}$.*

Furthermore, leveraging Pareto optimality (Zitzler & Thiele, 1999), we derive a corollary as follows:

**Corollary 2** *The optimal GIP-C solution is always better than that of the optimal ERM solution. For instance, $\mathcal{L}(\theta_{GIP\text{-}C}^*) > \mathcal{L}(\theta_{ERM}^*)$.*

Hence, GIP-C consistently exhibits superior performance compared with ERM, contributing as one of the most effective baselines to date. To determine the invariant gradient trajectory, we propose a meta-learning based approach (Finn et al., 2017) for our DG. Firstly, the agent aims to train its model parameter using domain-wise data via the optimization problem $\theta_i^{(r)} = \arg\min_\theta \mathcal{L}(\theta^{(r)}, S_i)$ at the local stage. Therefore, the domain-wise gradient trajectory can be computed via Eq. (4). At the meta update stage, the agent leverages the domain-wise gradients to approximate the invariant gradient trajectory $h_{\text{GIP-C}}$ using Theorem 1. Thereafter, the model is updated using updating function $\theta^{(r+1)} = \theta^{(r)} - \alpha h_{\text{GIP-C}}^{(r)}$. The detailed algorithm is presented in Alg. 1.

---

**Algorithm 1:** Domain Generalization via Pareto Optimality Gradient Matching

---

**Input** : Number of training domain $K$, initial model parameter $\theta^{(0)}$, learning rate $\alpha$.
**Output :** Model parameters $\theta$
1 **for** *each round $r = 0, \ldots, R$* **do**
2     **for** *domain $i \in \mathcal{K}$* **do**
3        Append the meta model to the domain-wise model $\theta_i^{(r,0)} \leftarrow \theta^{(r)}$.
4        **for** *local epoch $e \in E$* **do**
5           Sample mini-batch $\zeta$ from local data $S_i$
6           Update domain-wise model $\theta_i^{(r,e+1)} = \theta_i^{(r,e)} - \eta \nabla \mathcal{L}_i(\theta_i^{(r,e)}, \zeta)$.
7        **end for**
8     **end for**
9     Apply **Meta Update**
10 **end for**
11 **Meta Update**
12     Calculate $h_i^{(r)}$ from $\theta_i^{(r,E)}, \theta_i^{(r)}$ according to $h_i^{(r)} = \theta_i^{(r,E)} - \theta_i^{(r)}$.
13     Calculate $h_{\text{ERM}}^{(r)} = \frac{1}{K} \sum_{i=1}^K h_i^{(r)}$ as the average gradient update.
14     At the $r^{th}$ optimization, $\phi = \kappa^2 \|h_{\text{ERM}}^{(r)}\|^2$.
15     Find the optimal $\widetilde{\pi}$ set by solving $\widetilde{\pi} = \arg\min_\pi h_\pi^{(r)} \cdot h_{\text{ERM}}^{(r)} + \sqrt{\kappa} \|h_{\text{ERM}}^{(r)}\| \|h_\pi^{(r)}\|$, where
      $h_\pi^{(r)} = \sum_{i=1}^K \pi_i h_i^{(r)}, \pi_i \in [0, 1], \ \sum_{i=1}^K \pi_i = 1, \ \forall i \in \mathcal{K}$
16     Update
$$\theta^{(r+1)} = \theta^{(r)} - \alpha h_{\text{GIP-C}} \ \text{ where } h_{\text{GIP-C}} \text{ is defined via Theorem 1.}$$

---

### 4.3 Theoretical Analysis

We consider two theoretical analyses for our proposed method in accordance to gradient invariant and DG properties. For instance,

**Theorem 2 (Gradient Invariant Properties)** *Given $U_i = \nabla \mathcal{L}_i(\theta) \cdot \nabla \mathcal{L}_{GIP\text{-}C}(\theta)$ as the utility function of the Pareto Optimality problem. At each round $r$ where the Pareto is applied on different domains $i$, we have the GIP variance reduced accordingly to the number of learned epochs applied. For instance, $Var\left(U_i(\theta^{(r+1,e)})\right) \leq Var\left(U_i(\theta^{(r+1,e)})\right) \Big/ E^* \left(\frac{\eta^2 L}{2} - \eta\right)$.*

**Remark 2** *The number of epochs $E$ is related to the progress in gradient magnitude each round. Thus, GIP variance reduction also relates to the gradient variance $\left\|\nabla U_k(\theta^{(r,e)})\right\|^2$. As the gradient variance decreases, the gradient step becomes smaller, and thus the gradient invariant analysis becomes more stable.*

**Lemma 4 (Divergence of domains in the sources' convex hull (Albuquerque et al., 2021))** *Let* $d_\mathcal{V}[\mathcal{D}_i, \mathcal{D}_j] \leq \epsilon$, $\forall i, j \in [K]$. $\vartriangleleft_K$ *is the convex hull formed by source domains. The following inequality holds for the $\mathcal{V}$-divergence between any pair of domains $\mathcal{D}', \mathcal{D}'' \in \vartriangleleft_K$: $d_\mathcal{V}[\mathcal{D}', \mathcal{D}''] \leq \epsilon$.*

From these two lemmas, we have the risk bounds on DG:

**Theorem 3 (Optimal generalized risk bounds)** *If a target domain lies beneath the convex hull formed by $K$ source domains, then we can achieve the optimal generalized risk on target domains when the following holds:*

$$\mathcal{L}_L^* = \mathcal{L}_K^* + \frac{M}{2}\sqrt{\frac{1}{K^2}\sum_{i=1}^{K}\sum_{j=1}^{K} D_{KL}\Big[p_i(y|x,\theta^*)\|p_j(y|x,\theta^*)\Big] + \frac{1}{K^2}\sum_{i=1}^{K}\sum_{j=1}^{K} D_{KL}\Big[p_i(x|\theta^*)\|p_j(x|\theta^*)\Big]}$$

$$s.t. \quad \theta^* = \arg\max_\theta \sum_{\substack{i,j\in\mathcal{K} \\ i\neq j}} \nabla\mathcal{L}_i(\theta) \cdot \nabla\mathcal{L}_j(\theta), \tag{6}$$

*where $\mathcal{L}_K^*, \mathcal{L}_L^*$ are the source and target risks, respectively.*

Theorem 3 demonstrates that by maximizing the GIP among domain gradients during each meta update round, we can effectively reduce the generalization gap between the source and target domains.

**Remark 3** *The term $D_{KL}\Big[p_i(x|\theta^*)\|p_j(x|\theta^*)\Big]$ represents the distribution of the dataset between two domains $i, j$, and is independent of $\theta^*$. Therefore, we can consider this term irreducible.*

**Remark 4** *The term $D_{KL}\Big[p_i(y|x,\theta^*)\|p_j(y|x,\theta^*)\Big]$ refers to the gap between the two hypotheses $p_i(y|x,\theta^*), p_j(y|x,\theta^*)$. This term is optimizable, and in our work, we optimize it by finding the model $\theta^*$ that maximizes the similarity among domain-specific gradients.*

## 5 EXPERIMENTAL EVALUATION

### 5.1 ILLUSTRATIVE TOY TASKS

**Dataset and Settings Descriptions.** Based on Zhao et al. (2019), we introduced a synthetic binary classification dataset, named Rect-4, which comprises four distinct domains representing four different users. In each domain, a data sample $x_d = (x_{d,1}, x_{d,2})$ is randomly selected in the two-dimensional space with varying region distributions (see Appendix B for more details). To visualize the gradient of the toy dataset, we design a 1-layer, 2-parameter network. To this end, we can visualize the gradients in a 3-D space, consisting of 2 parameters and 1 weight.

**Results and Analysis.** Fig. 4 presents the performance of POGM on Rect-4 in the first 5 meta-training rounds. The results align with our empirical analysis in Sections 3.2 and 3.3. POGM demonstrates the smallest gradient divergence and the largest gradient update magnitude, indicating a closer progression towards the optimal solution. In contrast, Fish exhibits fluctuations across domains, resulting in a smaller meta-gradient magnitude and hindering convergence. Similarly, Fishr produces a large gradient divergence between domains. Despite the large magnitude of the domain-wise gradients, the small meta-gradient in Fishr prevents robust convergence. The problem of Fishr in the initial rounds will be discussed in detailed in Section 5.4.

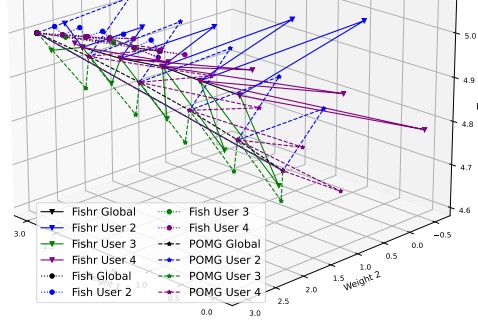

Figure 4: Illustrative toy task of POGM (optimal point is at $w_1 = 0, b = 0$).

Table 1: DomainBed benchmark. We format **first**, second, and worse than ERM results.

| Algorithm | Accuracy(↑) | | | | | | | | Ranking(↓) | | |
| | CMNIST | RMNIST | VLCS | PACS | OfficeHome | TerraInc | DomainNet | Avg | Arith. mean | Geom. mean | Median |
|---|---|---|---|---|---|---|---|---|---|---|---|
| ERM | 57.8 ± 0.2 | 97.8 ± 0.1 | 77.6 ± 0.3 | 86.7 ± 0.3 | 66.4 ± 0.5 | 53.0 ± 0.3 | 41.3 ± 0.1 | 68.7 | 9.00 | 8.0 | 10 |
| MTL (JMLR, 2021) | 57.6 ± 0.3 | 97.9 ± 0.1 | 77.7 ± 0.5 | 86.7 ± 0.2 | 66.5 ± 0.4 | 52.2 ± 0.4 | 40.8 ± 0.1 | 68.5 | 8.4 | 7.7 | 7 |
| SagNet (CVPR, 2021) | 58.2 ± 0.3 | 97.9 ± 0.0 | 77.6 ± 0.1 | 86.4 ± 0.4 | 67.5 ± 0.2 | 52.5 ± 0.4 | 40.8 ± 0.2 | 68.7 | 8.0 | 7.3 | 7 |
| ARM (NIPS, 2021) | 63.2 ± 0.7 | 98.1 ± 0.1 | 77.8 ± 0.3 | 85.8 ± 0.2 | 64.8 ± 0.4 | 51.2 ± 0.5 | 36.0 ± 0.2 | 68.1 | 10.0 | 8.3 | 10 |
| VREx (ICML, 2021) | 67.0 ± 1.3 | 97.9 ± 0.1 | 78.1 ± 0.2 | 87.2 ± 0.6 | 65.7 ± 0.3 | 51.4 ± 0.5 | 30.1 ± 3.7 | 68.2 | 8.1 | 6.1 | 7 |
| RSC (ECCV, 2020) | 58.5 ± 0.5 | 97.6 ± 0.1 | 77.8 ± 0.6 | 86.2 ± 0.5 | 66.5 ± 0.6 | 52.1 ± 0.2 | 38.9 ± 0.6 | 68.2 | 10.6 | 10.2 | 11 |
| AND-mask (ICLR, 2021) | 58.6 ± 0.4 | 97.5 ± 0.0 | 76.4 ± 0.4 | 86.4 ± 0.4 | 66.1 ± 0.2 | 49.8 ± 0.4 | 37.9 ± 0.6 | 67.5 | 13.0 | 12.7 | 13 |
| SAND-mask (ICML, 2021) | 62.3 ± 1.0 | 97.4 ± 0.1 | 76.2 ± 0.5 | 85.9 ± 0.4 | 65.9 ± 0.5 | 50.2 ± 0.1 | 32.2 ± 0.6 | 67.2 | 14.0 | 13.3 | 13 |
| EQRM (NIPS, 2022) | 53.4 ± 1.7 | 98.0 ± 0.0 | 77.8 ± 0.6 | 86.5 ± 0.2 | 67.5 ± 0.1 | 47.8 ± 0.6 | 41.0 ± 0.3 | 67.4 | 8.99 | 7.4 | 7 |
| RDM (WACV, 2024) | 57.5 ± 1.1 | 97.8 ± 0.0 | 78.4 ± 0.4 | 87.2 ± 0.7 | 67.3 ± 0.4 | 47.5 ± 1.0 | 41.4 ± 0.3 | 68.2 | 8.7 | 6.9 | 6 |
| SAGM (CVPR, 2023) | 63.4 ± 1.2 | 98.0 ± 0.1 | 79.9 ± 0.2 | 85.8 ± 0.8 | 65.3 ± 0.5 | 50.8 ± 0.6 | 38.5 ± 0.2 | 68.8 | 8.7 | 6.8 | 11 |
| CIRL (CVPR, 2022) | 62.1 ± 1.3 | 97.7 ± 0.2 | 78.6 ± 0.6 | 86.3 ± 0.4 | 67.1 ± 0.3 | 52.1 ± 0.2 | 39.7 ± 0.4 | 69.1 | 8.7 | 8.1 | 7 |
| MADG (NIPS, 2023) | 60.4 ± 0.8 | 97.9 ± 0.0 | 78.7 ± 0.2 | 86.5 ± 0.4 | **71.3** ± 0.5 | 53.7 ± 0.5 | 39.9 ± 0.2 | 69.8 | 5.1 | 4.0 | 5 |
| ITTA (CVPR, 2023) | 57.7 ± 0.6 | **98.5** ± 0.1 | 76.9 ± 0.6 | 83.8 ± 0.3 | 62.0 ± 0.2 | 43.2 ± 0.5 | 34.9 ± 0.1 | 65.3 | 14.1 | 10.9 | 16 |
| Mixstyle (ICLR, 2022) | 54.4 ± 1.4 | 97.9 ± 0.0 | 77.9 ± 0.1 | 85.2 ± 0.1 | 60.4 ± 0.5 | 44.0 ± 0.6 | 34.0 ± 0.3 | 64.8 | 14.0 | 12.8 | 17 |
| Fish (ICLR, 2022) | 61.8 ± 0.8 | 97.9 ± 0.1 | 77.8 ± 0.6 | 85.8 ± 0.6 | 66.0 ± 2.9 | 50.8 ± 0.4 | **43.4** ± 0.3 | 69.1 | 8.6 | 6.8 | 9 |
| Fishr (ICML, 2022) | **68.8** ± 1.4 | 97.8 ± 0.1 | 78.2 ± 0.2 | 86.9 ± 0.2 | 68.2 ± 0.2 | 53.6 ± 0.4 | 41.8 ± 0.2 | 70.8 | 4.6 | 3.6 | 3 |
| POGM (Ours) | 66.3 ± 1.2 | 97.9 ± 0.0 | **82.0** ± 0.1 | **88.4** ± 0.5 | 70.0 ± 0.3 | **54.8** ± 0.5 | 42.6 ± 0.2 | **71.7** | **2.1** | **1.8** | **2** |

## 5.2 RESULTS ON DOMAINBED BENCHMARK

We conducted extensive experiments to validate our proposed POGM on DomainBed (Gulrajani & Lopez-Paz, 2021). Our evaluation encompasses not only synthetic datasets like Colored MNIST (CMNIST) (Ghifary et al., 2015a) and Rotated MNIST (RMNIST) (Ghifary et al., 2015b), but also real-world multi-domain image classification datasets such as VLCS (Torralba & Efros, 2011), PACS (Li et al., 2017), OfficeHome (Venkateswara et al., 2017), Terra Incognita (Beery et al., 2018), and DomainNet (Peng et al., 2019). To ensure a fair comparison, we enforce strict training conditions. All methods are trained using only 20 different hyperparameter configurations and for the same number of steps. We then average the results over three trials. For a comprehensive understanding of our experimental setup, please refer to the detailed settings provided in Appendix A. Tab. 1 summarizes results on DomainBed using the "Test-domain" model selection: The validation set follows the same distribution as the test domain.

In DomainBed, ERM was carefully fine-tuned and therefore serves as a robust baseline. However, previous methods consistently fall short of achieving the highest scores across datasets. Specifically, invariant predictors like IRM and VREx, along with gradient masking approaches such as AND-mask, demonstrate poor performance on real datasets. Furthermore, CORAL not only underperforms compared to ERM on Terra Incognita but more crucially, it fails to identify correlation shifts on CMNIST. This is attributed to feature-based methods neglecting label considerations.

As observed from Tab. 1, our proposed method efficiently tackle correlation and diversity shifts. Besides the Fishr which shows the competitive results to our work, our POGM systematically performs better than ERM on all real datasets: the differences are over standard errors on VLCS (82.0% vs. 77.6%), PACS (88.4% vs. 86.7%), OfficeHome (70.0% vs. 66.4%), and on the large-scale Terra Inc. (54.8% vs. 53.0%), and DomainNet (42.6% vs. 41.3%). However, on synthetic data, our proposed POGM does not exhibit significant improvement over other baseline methods. Notably, POGM performs notably worse on CMNIST, which will be discussed further in the subsequent section. In summary, despite the decrease on synthetic datasets, POGM shows competitive results vs. Fish, and Fishr, outperforms the ERM on all challenging datasets. Significantly, POGM consistently achieves top results on real datasets, i.e., ranking in the top 1 and 2 positions on benchmark tests.

## 5.3 NUMERICAL STUDIES

### 5.3.1 INVARIANT GRADIENT PROPERTIES

Fig. 5 illustrates the invariant gradient properties of POGM. POGM shows a stronger correlation between two domain-specific gradient angles compared to ERM, Fish (refer to Fig. 3). Furthermore, every pair-wise gradient angle has smaller gap with each other, thus, the gradient of POGM shows better invariant properties than that of Fish and Fishr. The correlation implies that the angles of two gradient directions change at the same rate as the POGM, indicating invariant properties.

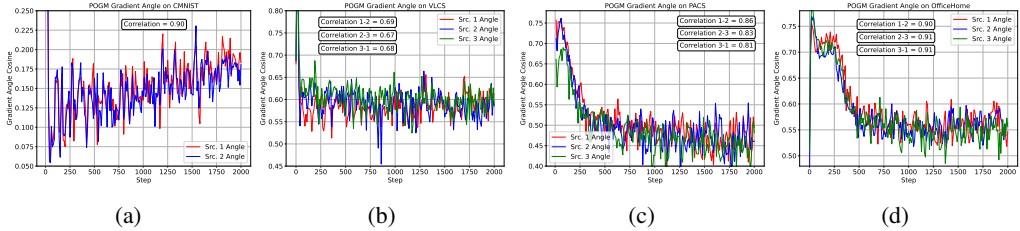

(a)         (b)         (c)         (d)

Figure 5: The correlation of gradient direction of POGM on CMNIST, VLCS, PACS, and OfficeHome. Compared to Fish and Fishr, POGM represents a higher correlation of gradient pairs among domains.

### 5.3.2 DOMAIN SPECIFIC GRADIENT ANGLE AND NORM DIFFERENCE

Besides the gradient invariant properties, we consider the average angle cosine with the domain-specific gradients $\cos(\theta_{\text{GIP-C}}^{(r)}, \theta_i^{(r)})$ and the norm distance with the domain-specific models $\|\theta_{\text{GIP-C}}^{(r)} - \theta_i^{(r)}\|$. If the gradients' angles and the norm distances are small, the generalization gap between the trained model and the domain-specific model is better, thus improving the validation set on the source domains. The results in Fig. 6 demonstrate that POGM outperforms Fish in retaining the performance on source domains when still improving the results on target domains. Additionally, models tend to perform better when the average gradient angle cosine exceeds 0, indicating most gradient conflicts (SHI et al., 2023) become insignificant (i.e., the angles become less than 90 degrees).

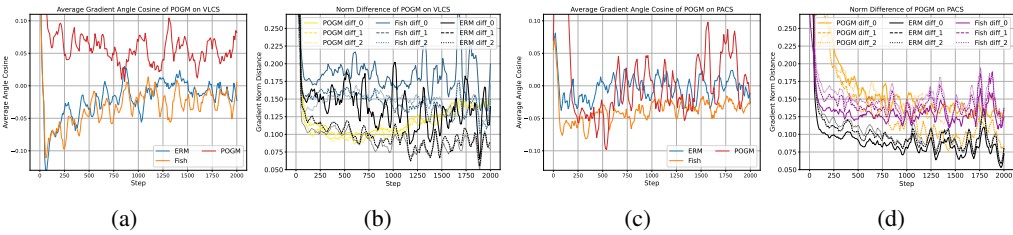

(a)         (b)         (c)         (d)

Figure 6: The gradients angles and the norm difference of ERM, Fish, Fishr, POGM on VLCS, PACS.

### 5.4 STABLE TRAINING BEHAVIOR OF POGM

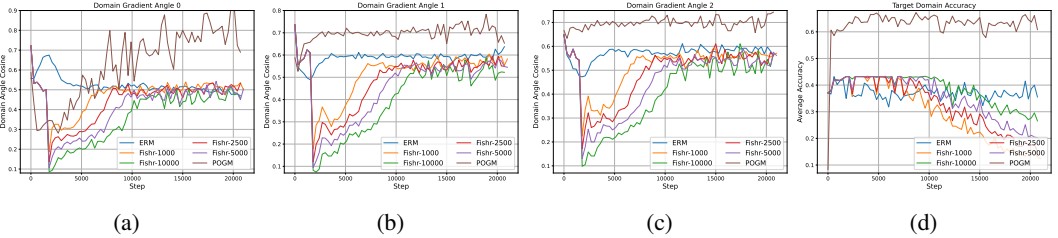

(a)         (b)         (c)         (d)

Figure 7: Invariant properties of POGM vs. ERM and Fishr.

Despite the robust results reported in the benchmarks, Fishr has drawback (which is mentioned in Section 3.2 and illustrated in Section 5.1) which impedes the convergence rate of Fishr in the initial rounds and can be further improved. Our introduced POGM is able to avoid the drawback of Fishr. Fig.7 illustrates the performance of POGM over Fishr and ERM in VLCS dataset. In Fishr, by minimizing the MSE among gradient variance and mean, the gradient angles of Fishr do not achieve the optimal results at the initial phases (from step 500 to 5000 in figures 7a, 7b, and 7c). In the latter stages, the Fishr angles become similar to that of ERM.

In contrast, by following GIP maximization proposed by (Shi et al., 2022), the angles between the learned gradient and other domains are optimized directly, resulting in a significant improvement in POGM over ERM and Fishr (see Fig. 7d).

## 5.5 INTEGRATABILITY

From Tab. 2, POGM demonstrates strong performance when combined with representation augmentation and mixing methods (e.g., Mixup and CIRL) in significant domain shifts (i.e., CM-NIST). This is because representation mixing reduces domain shifts among different domains (Yao et al., 2022).

Table 2: The integratibility of POGM

| Dataset | CMNIST | PACS | VLCS |
|---|---|---|---|
| POGM | $66.3 \pm 1.2$ | $88.4 \pm 0.5$ | $70.0 \pm 0.3$ |
| POGM + Mixup | $69.5 \pm 0.6$ | $89.1 \pm 0.6$ | $70.8 \pm 0.5$ |
| POGM + Data Aug. | $67.1 \pm 1.1$ | $88.6 \pm 0.5$ | $71.2 \pm 0.4$ |
| POGM + CIRL | $\mathbf{71.4} \pm 0.7$ | $90.1 \pm 0.4$ | $70.6 \pm 0.5$ |
| POGM + SWAD | $68.1 \pm 0.5$ | $\mathbf{91.2} \pm 0.3$ | $\mathbf{72.6} \pm 0.2$ |

Meanwhile, data augmentation expands the data samples, thereby increasing the effective domain sampling size and mitigating the negative impact of domain shifts. Additionally, POGM exhibits substantial robustness when integrated with sharpness-aware techniques, such as SWAD, in datasets with moderate domain shifts (e.g., PACS, VLCS). The reason for this is that sharpness-aware methods stabilize gradients, which in turn enhances gradient matching—an aspect that depends heavily on the stability of domain-specific gradients.

# 6 REDUCED COMPUTATIONAL OVERHEADS VIA HESSIAN-FREE APPROXIMATION

Fig. 8 demonstrates the computational efficiency in terms of the amount of GPU memory used and update time in each step, respectively. Firstly, POGM uses much less GPU memory than Fishr and ERM. This is because POGM does not need to approximate the Hessian, which saves memory during calculations. However, POGM uses more memory than Reptile-based Fish. This is because Reptile-based Fish trades off some performance for a simpler design, as explained in Section 5.2. Secondly, POGM is much faster than the other methods per step. This is because POGM only applies the meta-update every few rounds (while others apply every round), thereby significantly reducing the computation time.

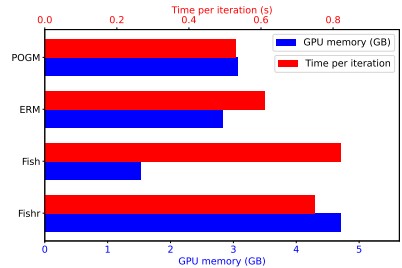

Figure 8: The evaluation of average GPU memory used during the training and the average time consumption per step when training Fish, Fishr, ERM, POGM with the VLCS dataset.

## 6.1 ABLATION STUDIES

Our ablation studies examine the impact of the number of domain-specific loop steps $E$, meta-update learning rate $\alpha$, the effects of selecting the hypersphere radius $\kappa$, and the impact of domain divergence on POGM. The results are shown in Appendix D.

# 7 CONCLUSION

In our paper, we tackled the challenge of out-of-distribution generalization. We conducted thorough empirical analyses on two state-of-the-art gradient-based methods, Fish (Shi et al., 2022) and Fishr (Rame et al., 2022), revealing two main issues: gradient magnitude elimination, and gradient fluctuation. These issues hinder both Fishr and Fish from consistently achieving peak performance. Building on these observations, we propose a novel approach that incorporates GIP from Fish and introduces a generalized regularization, called GIP-C, to ensure stability. We employ meta-learning to separate the domain-specific optimization stage from the GIP optimization phase, allowing for a Hessian-free approximation of our GIP-C optimization problem. Our experiments, which are reproducible with our open-source implementation, demonstrate that POGM delivers competitive results compared to Fishr and outperforms other baseline methods across various popular datasets. We anticipate that our learning architecture will pave the way for gradient-based out-of-distribution generalization without the need for Hessian approximation.

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

# A EXPERIMENTAL SETTINGS

## A.1 DATASETS

**Rotated MNIST** (Ghifary et al., 2015a) consists of 10000 digits in MNIST with different rotated angle $d$ such that each domain is determined by the degree $d \in \{0, 15, 30, 45, 60, 75\}$.

**Colored MNIST** (Gulrajani & Lopez-Paz, 2021) consists of 10000 digits in MNIST with different rotated angle $d$ such that each domain is determined by the different color set.

**PACS** (Li et al., 2017) includes 9991 images with 7 classes $y \in \{$dog, elephant, giraffe, guitar, horse, house, person$\}$ from 4 domains $d \in \{$art, cartoons, photos, sketches$\}$.

**VLCS** (Torralba & Efros, 2011) is composed of 10729 images, 5 classes $y \in \{$bird, car, chair, dog, person$\}$ from domains $d \in \{$Caltech101, LabelMe, SUN09, VOC2017$\}$.

**OfficeHome** (Venkateswara et al., 2017) includes 15500 images from 65 categories from four domains $d \in \{$Art, Clipart, Product, and Real-World$\}$ with various categories of objects commonly found in office and home settings. It's widely used for domain adaptation and generalization tasks.

**TerraIncognita** (Beery et al., 2018) contains 24330 images captured by satellites and aerial platforms, representing different terrains like forests, deserts, and urban areas.

**DomainNet** (Peng et al., 2019) is a large-scale dataset with 0.6 millions of images which are divided into 345 classes for DG research, featuring images from six domains $d \in \{$Clipart, Infograph, Painting, Quickdraw, Real, and Sketch$\}$.

## A.2 BASELINES

We compare POGM with the two most popular recent gradient-based DG, i.e., Fish (Shi et al., 2022), and Fishr (Rame et al., 2022) using GPUs: NVIDIA RTX 3090.

Besides, we also compare our POGM with other state-of-the-art methods, which are ERM, IRM (Wang et al., 2022), GroupDRO (Sagawa et al., 2020), Mixup (Yao et al., 2022), DANN (Sicilia et al., 2023), MTL (Blanchard et al., 2021), SagNet (Nam et al., 2021), ARM (Zhang et al., 2021), VREx (Krueger et al., 2021), RSC (Huang et al., 2022), AND-mask, SAND-mask (Shahtalebi et al., 2020), EQRM (Eastwood et al., 2022b), RDM (Nguyen et al., 2024), SAGM (Wang et al., 2023), CIRL (Lv et al., 2022), MADG (Dayal et al., 2023), ITTA (Chen et al., 2023), MixStyle (Zhou et al., 2021).

## A.3 IMPLEMENTATION DETAILS

We utilize different architectures for feature extraction and classification across datasets. Specifically, we employ a simple CNN for RMNIST and CMNIST, while ResNet-18 (He et al., 2016) is used for VLCS, PACS, and OfficeHome. For Terra Incognita and DomainNet, ResNet-50 (He et al., 2016) serves as the chosen architecture.

All experiments are trained for 100 epochs. During the local domain training phase, we employ SGD to optimize both the feature extractor and classifier. The initial learning rates are set to 0.001 for RMNIST and CMNIST, and 0.00005 for PACS, VLCS, Terra Incognita, OfficeHome, and DomainNet. Batch sizes are set to 64 for RMNIST and CMNIST, and 32 for the other datasets.

In the meta-learning phase, the meta-learning rate is set to 0.01, with the available searching hypersphere $\kappa$ set to 0.5. We conduct 5 local steps between each meta update.

Our code is based on DomainBed[1].

## A.4 HYPERPARAMETER SEARCH

Based on the experimental guidelines in (Gulrajani & Lopez-Paz, 2021), we perform a random search with 20 trials to fine-tune the hyperparameters for each algorithm and test domain. We divide the data from each domain into 80% for training, and evaluation; and 20% for selecting the hyperparameters.

---

[1]https://github.com/facebookresearch/DomainBed/tree/main/domainbed

To mitigate randomness, we experiment twice with different seeds. Finally, we present the average results from these repetitions along with their estimated standard error.

We perform a grid search over pre-defined values for each hyperparameter and report the optimal values along with the values used for the grid search. Furthermore, we do early stopping based on the validation accuracy on the source domain and use the models that obtain the best validation accuracy.

### A.5 MODEL SELECTION

In DG, choosing the right model is like a learning task itself. We adopt the test-domain validation method from (Gulrajani & Lopez-Paz, 2021), which is one of three selection methods. This approach is like consulting an oracle, as we pick the model that performs best on a validation set with the same distribution as the test domain.

## B TOY DATASET DESCRIPTION

We visualize the training data in Figs. 9 and 10. In the FDG setting, the users are from different domains. To this end, we design the data where the points are distributed into rectangular with different sizes and shapes. The rationale for designing the data distribution is as follows:

- The global dataset consists of two classes from two rectangular regions, which has the classification boundary equal to $y = 0$.
- Each domain-wise dataset has different classification boundary (e.g., $x = -6$ for domain 1). We add the noisy data on every domains so that the user assign to each domain will tend to learn the local boundary instead of the global boundary. Thus, we can observe the gradient divergence more clearly, as the global boundary is not the optimal solution when learn on local dataset.
- All of the local classification boundary is orthogonal from the global classification boundary, thus, we can make the learning more challenging despite the simplicity of the toy dataset.

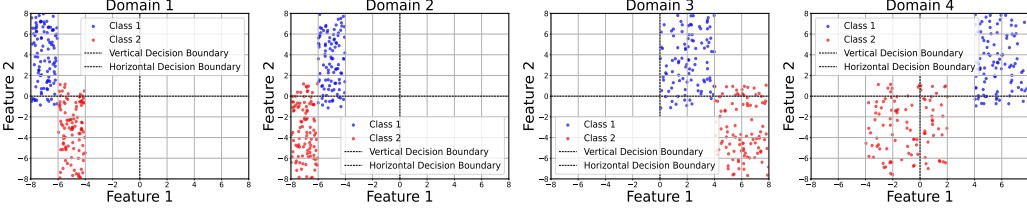

Figure 9: Illustration of users with different domains.

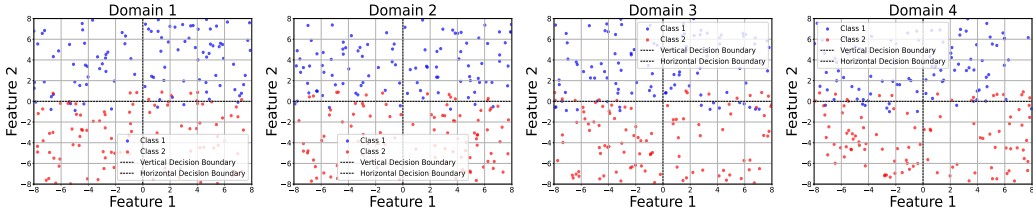

Figure 10: Illustration of users with same domains.

## C EMPIRICAL ANALYSIS

### C.1 DETAILED MEASUREMENT SETUP OF THE EMPIRICAL ANALYSIS

To evaluate the invariant gradient properties of Fish and Fishr, we conduct four following experiments.

### C.1.1 DOMAIN-SPECIFIC MODEL NORM DIFFERENCE

We conduct the domain-specific model norm difference to measure the empirical distance $\|\theta_i^{(r)} - \theta_{\Box}^{(r)}\|^2$ between domain-specific model $\theta_i^{(r)}$ and the learned model $\theta_{\Box}^{(r)}$, where $\Box = \{\text{Fish}, \text{Fishr}, \text{ERM}\}$ is the algorithms being evaluated.

To conduct the mentioned measurement, at each update round $r$, we conduct simultaneously two different training scenarios:

**Evaluated Algorithm Training.** From the previous model $\theta_{\Box}^{(r-1)}$, we train evaluated model $\theta_{\Box}^{(r)}$.

**Domain-specific Training.** From the previous model $\theta_{\Box}^{(r-1)}$, we train domain-specific model $\theta_i^{(r)}$ via the domain-specific data $S_i$.

By doing so, we only evaluate the divergence between the evaluated and domain-specific models at every round. Thus, our metric guarantees fairness among algorithms.

### C.1.2 DOMAIN-SPECIFIC GRADIENT ANGLE

We conduct the domain-specific model gradient angle to measure the cosine similarity $h_i^{(r)} \cdot h_{\Box}^{(r)}/\|h_i^{(r)}\|\|h_{\Box}^{(r)}\|$ between domain-specific gradient trajectory $h_i^{(r)}$ and the learned gradient trajectory $h_{\Box}^{(r)}$.

To conduct the mentioned measurement, at each update round $r$, we conduct simultaneously two different training scenarios:

**Evaluated Algorithm Training.** From the previous model $\theta_{\Box}^{(r-1)}$, we train evaluated model $\theta_{\Box}^{(r)}$. The evaluated gradient trajectory is measured as

$$h_{\Box}^{(r)} = \theta_{\Box}^{(r)} - \theta_{\Box}^{(r-1)}. \tag{7}$$

**Domain-specific Training.** From the previous model $\theta_{\Box}^{(r-1)}$, we train domain-specific model $\theta_i^{(r)}$ via the domain-specific data $S_i$. The domain-specific gradient trajectory is measured as

$$h_i^{(r)} = \theta_i^{(r)} - \theta_i^{(r-1)}. \tag{8}$$

By doing so, we only evaluate the divergence between the evaluated and domain-specific models at every round. Thus, our metric guarantees fairness among algorithms.

### C.1.3 INVARIANT GRADIENT ANGLE

To measure the invariant gradient angle, we first save a previously $\tau$-time trained model $\theta_{\Box}^{r-\tau}$. We measure the invariant gradient angle via the following cosine similarity formulation:

$$\text{Invariant Angle} = \frac{[\theta_{\Box}^{(r)} - \theta_{\Box}^{(r-1)}] \cdot [\theta_{\Box}^{(r)} - \theta_{\Box}^{(r-\tau)}]}{\left\|[\theta_{\Box}^{(r)} - \theta_{\Box}^{(r-1)}]\right\| \times \left\|[\theta_{\Box}^{(r)} - \theta_{\Box}^{(r-\tau)}]\right\|} \tag{9}$$

By doing so, we can approximate the fluctuation of the evaluated gradient. Specifically, as the gradient fluctuation issue is higher, the angle becomes larger, resulting in a smaller cosine similarity.

### C.2 GRADIENT MAGNITUDE NORM

We measure the gradient magnitude norm to evaluate how much the gradient made that round. As the gradient magnitude is larger, the gradient tends to progress more, and thus, make more impact on the learning progress. To measure the gradient magnitude norm, we calculate via the following formulation:

$$\text{Grad Norm} = \|\theta_{\Box}^{(r)} - \theta_{\Box}^{(r-1)}\|^2. \tag{10}$$

### C.3    EVALUATION ON GRADIENT ELIMINATION

In Fig. 2, the results are conducted from the real dataset VLCS, the gradient magnitude decreased from iteration 0 to 5000. This led to a notable decrease in the cosine of the gradient angle specific to the domain, as shown in Fig. 2a. Simultaneously, there was a significant reduction in the norm difference. This observation corresponds with the illustration in Fig. 1b, where the elimination of norm magnitude does not necessarily imply a reduction in gradient angle. This phenomenon results in the slower convergence of Fishr compared to that of the ERM, i.e., for nearly 5000 iterations (see Fig. 2d).

Fig. 2c demonstrates the behavior of regularization in Fishr, where the algorithm firstly aims to minimize the regularizer exhaustively, leading to the overfitting in the regularization. From iteration 7500 to 10000, the ERM loss in Fishr is considered instead of regularization minimization. And the Fishr only achieves the stable learning stage after the iteration of 10000.

When the Fishr goes into the stable learning stage, the gradient norm distance and gradient angle cosine become same with that of the ERM, resulting in the same performance between ERM and Fishr.

### C.4    EXTENSIVE ANALYSIS ON THE EFFECT OF DOMAIN DIVERGENCE TO THE POGM PERFORMANCE

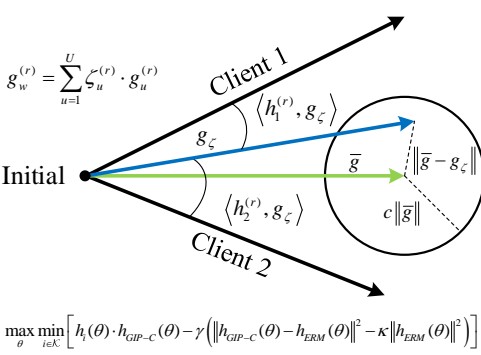

Figure 11

To detailed explain the effects of domain divergence to POGM performance, we first explain the rationale of POGM. As depicted in Fig. 11, the POGM gradients are learned via the maximization of $\sum_{i,j \in \mathcal{K}}^{i \neq j} \nabla \mathcal{L}_i(\theta) \cdot \nabla \mathcal{L}_j(\theta)$. Therefore, the learned POGM gradient vectors will be between all domain-wise gradient trajectories, which can be found in Fig. 11.

When training domains are diverse, the space between them is extensive, increasing the chance of test domains to be. This increases the likelihood that the learned POGM gradients will align well with test domains, especially those similar to certain training domains. Conversely, when training domains are similar, and especially when test domains differ significantly from them, problems arise.

**Lemma 5 (Domain Divergence)** *Given a set of gradients vectors $\mathcal{G}_{source} = \{g_1, \ldots, g_K\}$ get from the model $\theta$ when train in source domains $\mathcal{K} = \{D - i\}_{i=1}^{K}$. If there is a gradient vector $g_L$ created by target domain $L$ such that $g_L \cdot g_i < g_i \cdot g_j, \ \forall i, j \in K$, then $g_L$ resides outside the convex hull generated by the set $\mathcal{G}_{source}$.*

Lemma 5 shows that, as the target domains diverge from the training source domains, the learned gradient of POGM in particular and via GIP, in general, will suffer from the divergence with that of the gradients on test target domains. Thus, we believe that, to guarantee the performance of the POGM, the training domain must diverge (and suggested to diverge than that of the test domains). Therefore, the POGM can learn a more generalized characteristics of source and target domains.

# D ABLATION STUDIES

## D.1 DIFFERENT META UPDATE LEARNING RATE

### Table 3

| Model selection | $\alpha$ | CMNIST | RMNIST | VLCS | PACS | OfficeHome | TerraInc | DomainNet | Avg |
|---|---|---|---|---|---|---|---|---|---|
| Testing-domain | 0.01 | $65.2 \pm 0.9$ | $94.7 \pm 0.3$ | $81.3 \pm 0.3$ | $87.6 \pm 0.9$ | $68.7 \pm 0.5$ | $50.4 \pm 0.9$ | $30.3 \pm 2.0$ | 68.3 |
|  | 0.1 | $62.0 \pm 0.9$ | $93.4 \pm 0.9$ | $81.3 \pm 0.2$ | $88.5 \pm 0.5$ | $69.4 \pm 0.6$ | $51.4 \pm 0.5$ | $34.2 \pm 1.1$ | 68.6 |
|  | 0.5 | $64.7 \pm 1.2$ | $97.8 \pm 0.1$ | $80.5 \pm 0.4$ | $87.8 \pm 0.3$ | $68.6 \pm 0.4$ | $53.9 \pm 0.1$ | $40.0 \pm 0.4$ | 70.5 |
| Training-domain | 0.01 | $51.7 \pm 0.2$ | $94.0 \pm 0.2$ | $79.4 \pm 0.3$ | $82.8 \pm 0.7$ | $67.8 \pm 0.5$ | $46.0 \pm 1.1$ | $30.3 \pm 2.0$ | 64.6 |
|  | 0.1 | $51.1 \pm 0.2$ | $93.2 \pm 0.8$ | $79.7 \pm 0.6$ | $86.4 \pm 0.9$ | $69.2 \pm 0.7$ | $46.0 \pm 1.7$ | $34.2 \pm 1.1$ | 65.7 |
|  | 0.5 | $51.4 \pm 0.2$ | $97.7 \pm 0.1$ | $79.4 \pm 0.3$ | $85.8 \pm 0.6$ | $67.8 \pm 1.0$ | $46.5 \pm 0.4$ | $40.0 \pm 0.4$ | 66.9 |

The ablation test of meta update learning rate is demonstrated as in Tab. 3. On the synthetic dataset RMNIST, due to the high correlation among domains, the DG problem tends to be simple. Thus, by choosing a high learning rate, we can easily achieve the optimal state.

When dealing with the more challenging dataset (i.e., CMNIST with low correlation among domains, and real datasets such as VLCS, PACS, and OfficeHome) the low learning rate appears to be the efficient setting.

However, in real-world datasets with high dimensionality, and due to the large of the learning model (i.e., ResNet-50), the loss landscape exists significantly sharp minimizers. As a result, choosing a large meta-learning rate is efficient in these datasets.

## D.2 DIFFERENT DOMAIN-SPECIFIC TRAINING ITERATIONS

### Table 4

| Model selection | $E$ | CMNIST | RMNIST | VLCS | PACS | OfficeHome | TerraInc | DomainNet | Avg |
|---|---|---|---|---|---|---|---|---|---|
| Testing-domain | 1 | $63.8 \pm 1.2$ | $97.3 \pm 0.3$ | $79.7 \pm 0.6$ | $87.3 \pm 0.4$ | $67.5 \pm 0.4$ | $54.0 \pm 0.6$ | $40.0 \pm 0.4$ | 69.9 |
|  | 5 | $60.9 \pm 1.3$ | $95.8 \pm 0.7$ | $81.0 \pm 0.1$ | $87.5 \pm 0.9$ | $69.0 \pm 1.0$ | $49.9 \pm 1.0$ | $33.1 \pm 2.0$ | 68.2 |
|  | 10 | $60.3 \pm 1.2$ | $97.5 \pm 0.1$ | $81.5 \pm 0.4$ | $88.2 \pm 0.4$ | $69.7 \pm 0.2$ | $52.3 \pm 0.9$ | $32.1 \pm 0.6$ | 68.8 |
| Training-domain | 1 | $51.6 \pm 0.2$ | $97.2 \pm 0.3$ | $78.8 \pm 0.6$ | $84.0 \pm 0.3$ | $67.0 \pm 0.6$ | $46.0 \pm 0.4$ | $40.0 \pm 0.4$ | 66.4 |
|  | 5 | $51.3 \pm 0.1$ | $95.6 \pm 0.7$ | $80.2 \pm 0.3$ | $86.5 \pm 0.8$ | $68.7 \pm 1.0$ | $44.9 \pm 1.2$ | $33.1 \pm 2.0$ | 65.7 |
|  | 10 | $50.4 \pm 0.2$ | $97.3 \pm 0.2$ | $79.1 \pm 0.2$ | $86.0 \pm 0.3$ | $69.0 \pm 0.2$ | $47.1 \pm 0.9$ | $31.9 \pm 0.7$ | 65.8 |

Tab. 4 demonstrates the ablation test on different domain-specific training iterations on 7 evaluating datasets (i.e., RMNIST, CMNIST, VLCS, PACS, OfficeHome, Terra Incognita, DomainNet). The results differ depending on the dataset characteristics. For synthetic datasets like RMNIST and CMNIST, where the data is simpler, gradient trajectories are easily estimated. Thus, using a low number of domain-specific training iterations between each meta-update round (e.g., 1 round) yields optimal settings. However, for real-world datasets such as VLCS, PACS, OfficeHome, Terra Incognita, and DomainNet, employing a larger number of rounds leads to improved training outcomes.

## D.3 DIFFERENT SEARCHING HYPERSPHERE RADIUS

### Table 5

| Model selection | $\kappa$ | CMNIST | RMNIST | VLCS | PACS | OfficeHome | TerraInc | DomainNet | Avg |
|---|---|---|---|---|---|---|---|---|---|
| Testing-domain | 0.05 | $62.0 \pm 1.4$ | $96.3 \pm 0.5$ | $80.4 \pm 0.8$ | $85.7 \pm 1.5$ | $68.3 \pm 1.1$ | $49.9 \pm 0.9$ | $28.1 \pm 3.6$ | 67.2 |
|  | 0.1 | $64.5 \pm 1.6$ | $97.4 \pm 0.2$ | $81.2 \pm 0.4$ | $88.2 \pm 0.8$ | $68.7 \pm 0.6$ | $52.9 \pm 0.6$ | $31.6 \pm 0.9$ | 69.2 |
|  | 0.5 | $63.8 \pm 1.0$ | $97.2 \pm 0.3$ | $82.0 \pm 0.1$ | $88.3 \pm 0.4$ | $69.5 \pm 0.2$ | $53.9 \pm 0.1$ | $40.2 \pm 0.3$ | 70.7 |
| Training-domain | 0.05 | $51.4 \pm 0.1$ | $96.1 \pm 0.7$ | $79.0 \pm 0.8$ | $84.4 \pm 1.6$ | $68.1 \pm 1.3$ | $45.3 \pm 1.0$ | $28.1 \pm 3.6$ | 64.6 |
|  | 0.1 | $51.3 \pm 0.1$ | $97.2 \pm 0.3$ | $79.5 \pm 0.3$ | $84.7 \pm 0.8$ | $67.4 \pm 0.6$ | $45.4 \pm 1.5$ | $31.5 \pm 0.8$ | 65.3 |
|  | 0.5 | $51.0 \pm 0.5$ | $97.2 \pm 0.3$ | $78.7 \pm 0.3$ | $85.8 \pm 0.4$ | $68.6 \pm 0.4$ | $45.8 \pm 0.4$ | $40.1 \pm 0.3$ | 66.7 |

Tab. 5 demonstrates the ablation test on different searching hypersphere radius $\kappa$ on 7 evaluating datasets (i.e., RMNIST, CMNIST, VLCS, PACS, OfficeHome, Terra Incognita, DomainNet). Across almost all datasets, POGM performs well when the available search radius is set to $\kappa \geq 0.5$. We can explain this problem as follows. Because of significant gradient divergence across domains, the search space expands accordingly. Therefore, the optimal solution is significantly different from the

ERM solutions, likely due to gradient divergence, which expands the search space. Besides, in the synthetic dataset RMNIST, the difference among domains is not significant, which makes the search space become small. Therefore, the optimal solution is $\kappa = 0.1$.

# E    PROOF ON LEMMAS

## E.1    PROOF ON LEMMA 1

For a simple proof, we denote $g_i, g_j$ as the gradients trajectories of the model learned from different domains $\nabla \mathcal{L}_i(\theta), \nabla \mathcal{L}_j(\theta)$, respectively. We assume that each gradient $g_i, g_j, \forall i, j$ can be factorized as follows:

$$
\begin{aligned}
g_i &= k\widetilde{g}_i \\
g_j &= k\widetilde{g}_j,
\end{aligned}
\tag{11}
$$

where $k$ is the scaling factor, $\widetilde{g}_i, \widetilde{g}_j$ is the unit vector, and satisfies $\|\widetilde{g}_i\|, \|\widetilde{g}_j\| < \varepsilon, \forall \varepsilon$. Therefore, we have MSE loss from Fishr (Rame et al., 2022) of $\mathcal{L}_{\text{dist}}\left(\nabla \mathcal{L}_i(\theta), \nabla \mathcal{L}_j(\theta)\right)$ as follows:

$$
\|g_i - g_j\|^2 = \|k\widetilde{g}_i - k\widetilde{g}_j\|^2 = k^2 \|\widetilde{g}_i - \widetilde{g}_j\|^2 \leq k^2 \left(\|\widetilde{g}_i\|^2 + \|\widetilde{g}_j\|^2\right) \leq k^2 2\varepsilon.
\tag{12}
$$

Therefore, we have:

$$
\mathcal{L}_{\text{dist}}\left(g_i, g_j\right) = \frac{1}{K(K-1)} \sum_{i=1}^{K} \sum_{j \neq i}^{K} \|g_i - g_j\|^2 \leq k^2 2\varepsilon
\tag{13}
$$

From this case, it is obvious that when $k \to 0$, we have $\mathcal{L}_{\text{dist}}\left(g_i, g_j\right) < k^2 2\varepsilon$. Therefore, the regularization loss can achieve the minimal solution when $k \to 0$, which also means where $\|g_i\|, \|g_i\| \to 0$.

## E.2    PROOF ON LEMMA 2

We have

$$
\nabla \mathcal{L}_i(\theta) \cdot \nabla \mathcal{L}_{\text{GIP-C}} \geq \min_{i \in \mathcal{K}} \nabla \mathcal{L}_i(\theta) \cdot \nabla \mathcal{L}_{\text{GIP-C}}.
$$

Therefore, we have:

$$
\frac{1}{K} \sum_{i \in \mathcal{K}} \nabla \mathcal{L}_i(\theta) \cdot \nabla \mathcal{L}_{\text{GIP-C}} \geq \min_{i \in \mathcal{K}} \nabla \mathcal{L}_i(\theta) \cdot \nabla \mathcal{L}_{\text{GIP-C}}.
$$

## E.3    PROOF ON LEMMA 3

We have

$$
\theta^* = \arg\max_{\theta} \sum_{i \in \mathcal{K}} \mathcal{L}_i(\theta) \cdot \nabla \mathcal{L}_{\text{GIP-C}}(\theta) - \gamma \left(\|\nabla \mathcal{L}_{\text{GIP-C}}(\theta) - \nabla \mathcal{L}_{\text{ERM}}(\theta)\|^2 - \kappa \|\nabla \mathcal{L}_{\text{ERM}}(\theta)\|^2\right),
\tag{14}
$$

From Lemma 2 and definitions 1, 2, we have:

$$
\begin{aligned}
\mathcal{L}_{\text{avg}}(\theta^*) &= \sum_{i \in \mathcal{K}} \mathcal{L}_i(\theta^*) \cdot \nabla \mathcal{L}_{\text{GIP-C}}(\theta^*) - \gamma \left(\|\nabla \mathcal{L}_{\text{GIP-C}}(\theta^*) - \nabla \mathcal{L}_{\text{ERM}}(\theta^*)\|^2 - \kappa \|\nabla \mathcal{L}_{\text{ERM}}(\theta^*)\|^2\right) \\
&\geq \min_{i \in \mathcal{K}} \left[\mathcal{L}_i(\theta^*) \cdot \nabla \mathcal{L}_{\text{GIP-C}}(\theta^*) - \gamma \left(\|\nabla \mathcal{L}_{\text{GIP-C}}(\theta^*) - \nabla \mathcal{L}_{\text{ERM}}(\theta^*)\|^2 - \kappa \|\nabla \mathcal{L}_{\text{ERM}}(\theta^*)\|^2\right)\right] \\
&\geq \mathcal{L}_{\text{Pareto}}(\theta^*) = \mathcal{L}_{\text{Pareto}}(\theta^*).
\end{aligned}
\tag{15}
$$

Therefore, we have that $\theta^*$ is also the Pareto optimality solution.

### E.4    PROOF ON LEMMA 5

We define a convex hull $\mathcal{H}$ of the finite set $\{g_1, \ldots, g_K\}$:

$$\mathcal{H} = \{h^i, \ \forall i\} = \{\lambda_1 g_1 + \ldots + \lambda_K g_K \mid \lambda \succeq 0, \mathbf{1}^\top \lambda = 1\}, \tag{16}$$

it is obvious that to guarantee the point to be in the convex hull, we must satisfy the condition: $\lambda_k < 1, \ \forall k$. Consider the target gradient vector $g_L$, where we have $g_L \cdot g_i < g_i \cdot g_j, \ \forall i, j \in K$. To prove that $g_L$ is outside of the convex hull, we leverage a contradict consumption, where $g_L$ is in the convex hull, and $g_L = \lambda_1 g_1 + \ldots + \lambda_K g_K$. Then, we have

$$\begin{aligned} g_L \cdot g_i &= (\lambda_1 g_1 + \ldots + \lambda_K g_K) \cdot g_i \\ &= \lambda_1 (g_1 \cdot g_i) + \ldots + \lambda_K (g_K \cdot g_i) \\ &< (\lambda_1 + \ldots + \lambda_K)(g_L \cdot g_i) = g_L \cdot g_i \end{aligned} \tag{17}$$

This does not exist, thus we have $g_L$ outside of the convex hull $\mathcal{H}$.

### E.5    PROOF ON LEMMA 6

The update of the model can be written as follows:

$$\theta^{(r,e+1)} = \theta^{(r,e)} - \eta \nabla U_k(\theta^{(r,e)}). \tag{18}$$

Now using the Lipschitz-smoothness assumption, we have

$$\begin{aligned} U_k(\theta^{(r,e+1)}) - U_k(\theta^{(r,e)}) &\leq -\eta \Big\langle \nabla U_k(\theta^{(r,e)}), \nabla U_k(\theta^{(r,e)}) \Big\rangle + \frac{\eta^2 L}{2} \Big\| \nabla U_k(\theta^{(r,e)}) \Big\|^2 \\ &= \Big( \frac{\eta^2 L}{2} - \eta \Big) \Big\| \nabla U_k(\theta^{(r,e)}) \Big\|^2. \end{aligned} \tag{19}$$

Averaging over all rounds, we have

$$U_k(\theta^{(r,e+1)}) - U_k(\theta^{(r,0)}) = \Big( \frac{\eta^2 L}{2} - \eta \Big) \sum_{e=0}^{E} \Big\| \nabla U_k(\theta^{(r,e)}) \Big\|^2. \tag{20}$$

Rearranging terms, we have

$$\Big\| \nabla U_k(\theta^{(r,e)}) \Big\|^2 \leq \frac{U_k(\theta^{(r,E^*)}) - U_k(\theta^{(r,0)})}{E^* \Big( \frac{\eta^2 L}{2} - \eta \Big)}. \tag{21}$$

## F    PROOF ON THEOREMS

### F.1    PROOF ON THEOREM 1

**Theorem 4 (Invariant Gradient Solution)** *Given the Pareto condition as mentioned in Lemma 3, $\widetilde{\pi} = \{\pi_1^{(r)}, \ldots, \pi_K^{(r)}\}$ are the set of $K$ learnable scaling parameters for the joint learner at each $r$ communication round. The invariant gradient $h_{GIP\text{-}C}$ is characterized by the*

$$h_{GIP\text{-}C}^{(r)} = h_{ERM}^{(r)} + \frac{\kappa \|h_{ERM}^{(r)}\|}{\|h_\pi^{(r)}\|} h_\pi^{(r)} \quad s.t. \quad \widetilde{\pi} = \arg\min_\pi h_\pi^{(r)} \cdot h_{ERM}^{(r)} + \kappa \|h_{ERM}^{(r)}\| \|h_\pi^{(r)}\| \tag{22}$$

*where $h_\pi^{(r)} = \sum_{i=1}^{K} \pi_i^{(r)} h_i^{(r)}$.*

*Proof.* For a clear proof, we denote $h_{GIP\text{-}C}^{(r)} = h_{GIP\text{-}C}^{(r)}(\theta)$, $h_{ERM}^{(r)} = h_{ERM}^{(r)}(\theta)$, $h_i^{(r)} = h_i^{(r)}(\theta)$, and $h_\pi^{(r)} = h_\pi^{(r)}(\theta)$. Therefore, we have

$$\max_\theta \min_{i \in \mathcal{K}} \Big[ h_i^{(r)}(\theta) \cdot h_{GIP\text{-}C}^{(r)}(\theta) - \gamma \Big( \|h_{GIP\text{-}C}^{(r)}(\theta) - h_{ERM}^{(r)}(\theta)\|^2 - \kappa \|h_{ERM}^{(r)}(\theta)\|^2 \Big) \Big]. \tag{23}$$

This is equivalent to

$$\max_\theta \min_\pi \left[ h_\pi^{(r)}(\theta) \cdot h_{\text{GIP-C}}^{(r)}(\theta) - \gamma \left( \|h_{\text{GIP-C}}^{(r)}(\theta) - h_{\text{ERM}}^{(r)}(\theta)\|^2 - \kappa \|h_{\text{ERM}}^{(r)}(\theta)\|^2 \right) \right], \qquad (24)$$

We first deal with the following minimization problem

$$\min_\pi U(\pi) = h_\pi^{(r)}(\theta) \cdot h_{\text{GIP-C}}^{(r)}(\theta) - \gamma \left( \|h_{\text{GIP-C}}^{(r)}(\theta) - h_{\text{ERM}}^{(r)}(\theta)\|^2 - \kappa \|h_{\text{ERM}}^{(r)}(\theta)\|^2 \right). \qquad (25)$$

To find the relaxation of the minimization $U(\pi)$, we fix $\pi, \gamma$ to find the optimal state of $h_{\text{GIP-C}}^{(r)}(\theta)$. The minimization is achieved when $\nabla_{h_{\text{GIP-C}}^{(r)}(\theta)} U(\pi) = 0$. For instance, we have:

$$\nabla_{h_{\text{GIP-C}}^{(r)}(\theta)} U(\pi) = h_\pi^{(r)}(\theta) - 2\gamma \left( h_{\text{GIP-C}}^{(r)}(\theta) - h_{\text{ERM}}^{(r)}(\theta) \right) \nabla_{h_{\text{GIP-C}}^{(r)}(\theta)} (h_{\text{GIP-C}}^{(r)}(\theta))$$

$$= h_\pi^{(r)}(\theta) - 2\gamma \left( h_{\text{GIP-C}}^{(r)}(\theta) - h_{\text{ERM}}^{(r)}(\theta) \right) = 0. \qquad (26)$$

This equality is achieved when

$$h_{\text{GIP-C}}^{(r)}(\theta) = h_{\text{ERM}}^{(r)}(\theta) + \frac{h_\pi^{(r)}(\theta)}{2\gamma}. \qquad (27)$$

Replace the solution found in Eq. (27) with Eq. (25), we have:

$$U(\pi) = h_\pi^{(r)}(\theta) \cdot \left[ h_{\text{ERM}}^{(r)}(\theta) + \frac{h_\pi^{(r)}(\theta)}{2\gamma} \right] - \gamma \left( \|h_{\text{GIP-C}}^{(r)}(\theta) - h_{\text{ERM}}^{(r)}(\theta)\|^2 - \kappa \|h_{\text{ERM}}^{(r)}(\theta)\|^2 \right)$$

$$= h_\pi^{(r)}(\theta) \cdot h_{\text{ERM}}^{(r)}(\theta) + h_\pi^{(r)}(\theta) \cdot \frac{h_\pi^{(r)}(\theta)}{2\gamma} - \gamma \left( \|h_{\text{GIP-C}}^{(r)}(\theta) - h_{\text{ERM}}^{(r)}(\theta)\|^2 - \kappa \|h_{\text{ERM}}^{(r)}(\theta)\|^2 \right)$$

$$= h_\pi^{(r)}(\theta) \cdot h_{\text{ERM}}^{(r)}(\theta) + \frac{1}{2\gamma} \|h_\pi^{(r)}(\theta)\|^2 - \gamma \left( \|\frac{h_\pi^{(r)}(\theta)}{2\gamma}\|^2 - \kappa \|h_{\text{ERM}}^{(r)}(\theta)\|^2 \right)$$

$$= h_\pi^{(r)}(\theta) \cdot h_{\text{ERM}}^{(r)}(\theta) + \frac{1}{2\gamma} \|h_\pi^{(r)}(\theta)\|^2 - \frac{1}{4\gamma} \|h_\pi^{(r)}(\theta)\|^2 + \gamma\kappa \|h_{\text{ERM}}^{(r)}(\theta)\|^2$$

$$= h_\pi^{(r)}(\theta) \cdot h_{\text{ERM}}^{(r)}(\theta) + \frac{1}{4\gamma} \|h_\pi^{(r)}(\theta)\|^2 + \gamma\kappa \|h_{\text{ERM}}^{(r)}(\theta)\|^2. \qquad (28)$$

By fixing the $\gamma$, we have:

$$\nabla_\gamma U(\pi) = -\frac{1}{4\gamma^2} \|h_\pi^{(r)}(\theta)\|^2 + \kappa \|h_{\text{ERM}}^{(r)}(\theta)\|^2 = 0. \qquad (29)$$

the optimal solution is when $\gamma$ satisfies:

$$\gamma = \sqrt{\frac{1}{4} \|h_\pi^{(r)}(\theta)\|^2 \Big/ \kappa \|h_{\text{ERM}}^{(r)}(\theta)\|^2} = \frac{\|h_\pi^{(r)}(\theta)\|}{2\sqrt{\kappa} \|h_{\text{ERM}}^{(r)}(\theta)\|}. \qquad (30)$$

Replace $\gamma$ into Eq. (28), we have:

$$U(\pi) = h_\pi^{(r)}(\theta) \cdot h_{\text{ERM}}^{(r)}(\theta) + \sqrt{\kappa} \|h_\pi^{(r)}(\theta)\| \|h_{\text{ERM}}^{(r)}(\theta)\|. \qquad (31)$$

### F.2 PROOF ON THEOREM 2

Consider the GIP optimization problem from Eq. (2), taking the GIP value as $U_i = g_{\text{GIP-C}} \cdot g_i$, we consider the variance of the GIP function at each round $r$ (when each specific domain $k$ are being chosen to optimized) as:

$$\text{Var}\left( U_i(\theta^{(r+1,e)}) \right) = \sum^{i \in K} U_i(\theta^{(r,e)}) - \max \min U_i(\theta^{(r,e)})$$

$$= \frac{K-1}{K} \left( \sum_{\substack{i \neq k}}^{i \in K} U_i(\theta^{(r,e)}) - \max_\theta U_k(\theta^{(r,e)}) \right). \qquad (32)$$

After each step, only the $U_k$ is updated via maximization problem, i.e., $\theta^{(e+1)} = \theta^{(e)} - \eta \nabla U_k(\theta^{(e)})$. Then, we have:

$$U_k(\theta^{(r,e+1)}) = U_k\Big(\theta^{(r,e)} - \eta \nabla U_k(\theta^{(r,e)})\Big) = U_k\Big(\theta^{(r,e)}\Big) - \eta\Big[\nabla U_k(\theta^{(r,e)})\Big]^2. \qquad (33)$$

Therefore, we have the Eq. (32) as follows:

$$\mathrm{Var}\Big(U_i(\theta^{(r+1,e)})\Big) = \frac{K-1}{K}\Big(\sum_{\substack{i \neq k}}^{i \in K} U_i - U_k + \eta\Big[\nabla U_k(\theta^{(r,e)})\Big]^2\Big). \qquad (34)$$

It is obvious that when $U_k(\theta^{(e+1)})$ conduct 1 step and in the next step, the different domain $k'$ is being chosen as Pareto fronts, we have the following:

$$\mathrm{Var}\Big(U_i(\theta^{(r+1,e)})\Big) \leq \eta\Big\|\nabla U_k(\theta^{(r,e*)})\Big\|^2. \qquad (35)$$

To bound the Eq. (35), we have the following lemma:

**Lemma 6** *The gradient variance norm after $E^*$ rounds can be considered as*

$$\Big\|\nabla U_k(\theta^{(r,e)})\Big\|^2 \leq \frac{U_k(\theta^{(r,E^*)}) - U_k(\theta^{(r,0)})}{E^*\Big(\frac{\eta^2 L}{2} - \eta\Big)}. \qquad (36)$$

Therefore, we have:

$$\mathrm{Var}\Big(U_i(\theta^{(r+1,e)})\Big) \leq \frac{U_k(\theta^{(r,E^*)}) - U_k(\theta^{(r,0)})}{E^*\Big(\frac{\eta^2 L}{2} - \eta\Big)} = \frac{\mathrm{Var}\Big(U_i(\theta^{(r+1,e)})\Big)}{E^*\Big(\frac{\eta^2 L}{2} - \eta\Big)}. \qquad (37)$$

### F.3 PROOF ON THEOREM 3

From (Nguyen et al., 2022), we have

$$\mathcal{L}_{\mathrm{test}} \leq \mathcal{L}_{\mathrm{train}} + \frac{M}{2}\sqrt{\frac{1}{K}\sum_{i=1}^{K} D_{\mathrm{KL}}\Big[p_T(y|\theta)\|p_i(y|\theta)\Big]}. \qquad (38)$$

From Lemma 4, we have:

$$\mathcal{L}_{\mathrm{test}} \leq \mathcal{L}_{\mathrm{train}} + \frac{M}{2}\sqrt{\frac{1}{K^2}\sum_{i=1}^{K}\sum_{j=1}^{K} D_{\mathrm{KL}}\Big[p_i(y|\theta)\|p_j(y|\theta)\Big]}$$

$$= \mathcal{L}_{\mathrm{train}} + \frac{M}{2}\sqrt{\frac{1}{K^2}\sum_{i=1}^{K}\sum_{j=1}^{K} D_{\mathrm{KL}}\Big[p_i(y|x,\theta)p_i(x|\theta)\|p_j(y|x,\theta)p_j(x|\theta)\Big]}$$

$$= \mathcal{L}_{\mathrm{train}} + \frac{M}{2}\sqrt{\underbrace{\frac{1}{K^2}\sum_{i=1}^{K}\sum_{j=1}^{K} D_{\mathrm{KL}}\Big[p_i(y|x,\theta)\|p_j(y|x,\theta)\Big]}_{\mathrm{B1}} + \underbrace{\frac{1}{K^2}\sum_{i=1}^{K}\sum_{j=1}^{K} D_{\mathrm{KL}}\Big[p_i(x|\theta)\|p_j(x|\theta)\Big]}_{\mathrm{B2}}}, \qquad (39)$$

where B1 is the divergence of the inference of model $\theta$ on different domains $i, j$. B2 is the divergence between domains, which is not tunable. This also means that, as the GIP $g_i \cdot g_j$ is maximized, we can improve the model generalization as follows:

$$\mathcal{L}_{\mathrm{test}}^* = \mathcal{L}_{\mathrm{train}}^* + \frac{M}{2}\sqrt{\frac{1}{K^2}\sum_{i=1}^{K}\sum_{j=1}^{K} D_{\mathrm{KL}}\Big[p_i(y|x,\theta^*)\|p_j(y|x,\theta^*)\Big] + \frac{1}{K^2}\sum_{i=1}^{K}\sum_{j=1}^{K} D_{\mathrm{KL}}\Big[p_i(x|\theta^*)\|p_j(x|\theta^*)\Big]}$$

$$\mathrm{s.t.} \quad \theta^* = \arg\max_{\theta} \sum_{\substack{i,j \in \mathcal{K}}}^{i \neq j} \nabla \mathcal{L}_i(\theta) \cdot \nabla \mathcal{L}_j(\theta). \qquad (40)$$

