# OpenReview forum: "DOMAIN GENERALIZATION VIA PARETO OPTIMAL GRADIENT MATCHING"
_ICLR.cc/2025/Conference — Submitted to ICLR 2025_

### Official Review · Reviewer_qp1y · 2024-11-01

**Soundness:** 2
**Presentation:** 2
**Contribution:** 2
**Rating:** 3
**Confidence:** 4

**Summary:**

This paper propose POGM (Pareto Optimality Gradient Matching), a new domain generalization method. POGM builds upon Fish and Fishr and improves their gradient fluctuations and gradient magnitude elimination issues. The method leverages Pareto optimality to reduce computational complexity and employs meta-learning to avoid expensive Hessian approximations. The results on seven datasets from Domainbed demonstrated that POGM achieves slightly better performance than selected baselines. The approach is supported by theoretical analysis, including proofs of gradient invariant properties and optimal generalized risk bounds.

**Strengths:**

1. The paper builds upon Fish and Fishr, and learn a new update direction, which is named as invariant gradient hGIP-C to  optimize the model parameters.
2. The approach is supported by theoretical analysis, including proofs of gradient invariant properties and optimal generalized risk bounds.
3. The authors use seven datasets form the Domainbed suit to demonstrate the effectiveness of the POGM.
4. The paper uses illustrative simulated examples to support their motivations and insights.

**Weaknesses:**

1. The paper is missing relevant baselines. POGM is learning a new gradient direction by restricting the searching space within a κ-hypersphere, which has the center determined by the ERM trajectory, see line 16 in Alg.1. [1] is improving the gradient matching by learning gradient direction applying PCA on the optimization trajectory, whose center is also determined by the ERM trajectory. Please compare and analyze the difference for their closer relevance.

2. POGM is building upon ERM. There are domain specific parameters as in line 6 in Alg1. There are also shared parameters as in line 3 in alg1. You have domain specific update and meta update.  How could its time per iteration less than ERM in Table 8. Why the memory usage is comparable to ERM?

3. I am not convinced by Lemma 1. If a flat region is shared across all training domains (i.e. the gradient of all domain specific loss w.r.t \theta is 0), it usually guarantees a better generalization ability. Because you have no prior about the testing domain. Many DG papers are building upon the assumption. To list a few [2][3][4]

4. Fig4 is trying to say POMG enables update with smoother trajectories. However, I do not think it is sufficient to derive the conclusion as (1) it is really hard to read. (2) Why are there four different users? The goal is to learn a single model with high DG ability. Can you please explain what different users mean here?

5. Fig6’s caption says it visualizes the gradients angles and the norm difference of ERM, Fish, Fishr, POGM on VLCS, PACS. I didn’t see any results from Fishr.

6. The meta update $\alpha$ is important factor to consider. Table 3 in the appendix presents the results for $\alpha$ = [0.01, 0.1, 0.5] The results show that increase alpha will always give better DG results. Why not keep increasing it? In summary, the comparison range is too limited. No conclusion can be derived from it.

[1] Pgrad: Learning principal gradients for domain generalization

[2] SWAD: Domain Generalization by Seeking Flat Minima

[3]  Invariant Risk Minimization

[4] Flatness-Aware Minimization for Domain Generalization

**Questions:**

See my weakness list above

---

### Official Review · Reviewer_jgxy · 2024-11-02

**Soundness:** 2
**Presentation:** 1
**Contribution:** 2
**Rating:** 5
**Confidence:** 4

**Summary:**

In the Domain Generalization problem, to better address the gradient conflict of different domains during gradient matching (e.g., Fishr), the paper proposes to employ the Pareto Optimality theory to find the optimal invariant updating direction. Instead of adopting the second-order computation, the paper uses meta-learning strategy for optimization. The experiments on domainbed shows its effectiveness.

**Strengths:**

1. The experiments of the proposed method achieve surprisingly good performance than previous methods.

**Weaknesses:**

1. The paper is poorly written and hard to follow, especially in Sec. 4 where it is hard to understand the logic of method design.
2. The Pareto optimality is employed only for the worst-case matching gradient, which is actually unnecessary. Maybe the combination of $\pi$ is suitable to apply Pareto optimality.
3. Previous related methods that adopt Pareto Optimality [1,2], and training trajectory [3] should be discussed and compared.
4. Why employ the trajectory matching and how it works is not clearly discussed.

[1] Chen, Yongqiang, et al. "Pareto invariant risk minimization: Towards mitigating the optimization dilemma in out-of-distribution generalization." arXiv preprint arXiv:2206.07766 (2022).
[2] Ye, Mao, and Qiang Liu. "Pareto navigation gradient descent: a first-order algorithm for optimization in pareto set." Uncertainty in Artificial Intelligence. PMLR, 2022.
[3] Zhang, Jian, et al. "Mvdg: A unified multi-view framework for domain generalization." European Conference on Computer Vision. Cham: Springer Nature Switzerland, 2022.

**Questions:**

See Weakness.

---

### Official Review · Reviewer_oWps · 2024-11-03

**Soundness:** 2
**Presentation:** 3
**Contribution:** 2
**Rating:** 5
**Confidence:** 3

**Summary:**

This work addresses two challenges in existing gradient-based domain generalization (DG) methods: gradient fluctuations and the elimination of gradient magnitudes. To tackle these issues, the paper proposes a Pareto Optimality Gradient Matching (POGM) method. POGM mitigates the problems of magnitude elimination and fluctuations by reformulating gradient matching through the summation of pairs of gradient inner products (GIPs) and restricting the search space for the GIP solution within a κ-hypersphere centered on the empirical risk minimization (ERM) gradient trajectory.

**Strengths:**

1. The proposed POGM method is well-motivated, with sufficient theoretical and empirical analysis on existing gradient-based DG methods.
2. The paper is well-written and the methodology is clearly explained .
3. The proposed POGM method is evaluated on several benchmark DG datasets.

**Weaknesses:**

1. Missing comparison with other baselines based on training-domain selection method. As mentioned in DomainBed (Gulrajani & Lopez-Paz, 2021), training-domain selection or leave-one-out method is better suited for the DG setting.
2. As shown in Table 1, POGM achieves only marginal improvements compared to other baseline methods. Specifically, it achieves the best performance on only  3 out of the 7 datasets.
3. The base feature extractor used in POGM differs for the VLCS, PACS, and OfficeHome datasets compared to DomainBed baselines. POGM uses ResNet18, whereas DomainBed uses ResNet50. Did the authors re-evaluate all the baselines listed in Table 1 accordingly for a fair comparison?
4. Missing hyperparameter search values for POGM-specific hyperparameters. The authors need to provide the list of hyperparameters and their search space.
5. Figure 8 is not explained clearly. It is mentioned in the text that POGM uses less GPU memory than ERM; however, the figure shows the opposite. Additionally, more explanation is needed regarding why the time taken for ERM is greater than for POGM, as ERM does not involve any Hessian calculation or gradient matching.

**Questions:**

Please see weaknesses.

---

### Official Review · Reviewer_oDrx · 2024-11-05

**Soundness:** 3
**Presentation:** 2
**Contribution:** 2
**Rating:** 5
**Confidence:** 4

**Summary:**

This work develops an algorithm for gradient-based domain generalisation that seeks to address particular issues arising from prior work tackling the same problem. In the case of Fish [Shi et al., 2022], the claim is that the IGDM measure, through the Fish algorithm, ends up having issues with the gradient direction fluctuating. In the case of Fishr [Rame et al., 2022], the claim is that mean-squared error employed to reduce angle between gradients can have a degenerate solition when the magnitute of the two gradients are zero.
To handle this, an new algorithm termed pareto optimal gradient matching (POGM) is developed, employed a meta-learning step to constrain the typical GIP-derived gradient from straying too far from the standard ERM gradient.
The utility of this method is first demonstrated on a toy setting before full evaluation on the DomainBed benchmarks.

**Strengths:**

The idea of effectively constraining or regularising the DG-targeted GIP by ERM seems novel and interesting, and the results appear to show that things improve (modulo weaknesses discussed below).
Moreover, the meta-learning setup seems to additionally allow for efficiencies in terms of choosing when to optimise the meta learner itself, rather than all the time. The experiments seem quite thorough also with well-standardised settings and hyperparameter search.

**Weaknesses:**

The main weaknesses with this work are the following:

a. The discussion of the shotcomings of Fish and Fishr are quite difficult to make sense of. For a piece of work that relies on these for motivating a fix, this should be done much better than it currently is, where it seems quite rushed and disjointed.

   1. Figure 1 has a lot of stuff in it that isn't explained in the captions.
      What is the reader to make of the arrows and trajectory of the optimisation in (1a) with regard to gradient fluctuations?
      What are the hatted version of $$g$$ in (1b), and what is the equation trying to say about gradient magnitudes being eliminated?

   2. The text and plots describing the issues (Sec 3.2 & 3.3) are incredibly terse and don't explain things in the figure. E.g. what is 'Fishr_10000' vs 'Fishr_2500'? What do subplots (2c) and (2d) have to do with the magnitude elimination?
      Similarly for Fish, the text claims that Fish has lower correlation among domain-specific gradients than ERM, but it's not clear what the training dynamics over steps shown in Fig (3) is supposed to convey? Is there a trend to identify here? The overall correlation is given in a (hard to read) box on the plot which seems relevant, but is the rest?
      Beyond this, the implications for issues drawn also appear to be demonstrated on a single instance---the plots show a single run of the model training. Does this hold across multiple runs?
      Is the correlation shown in the plots computed across the steps in a single run, or across both steps and runs for multiple runs?
      Without a clear and sound demonstration of the actual issues claimed in the prior work, it's hard to judge merit of this work. While Lemma 1 is clearly true, it needs to be shown much more clearly with statistics that it actually has an effect.

  3. The formulation of the GIP-C objective could also be clearer.
     Eq (1) implies that the objective is a function of both $\theta$ and the set of domains $\mathcal{K}$, and is recursive in $\mathcal{L}_{GIP-C}$ on account of its gradient.
     Eq (2) replaces the search over all pairs (i, j) with each domain to the GIP-C gradient (i, GIP-C); however, does this replacement not have an effect on the gradient of $\mathcal{L}_{GIP-C}$ in the second term? If this effect is to be discounted, it is not clear why/how.
     This also affects the lower bound formulation in Lemma 3 where the inequality in Lemma 2 applies for the _first term_ in Eq (1), but it's not clear why this wouldn't affect the second term as both $\mathcal{L}_{ERM}$ and $\mathcal{L}_{GIP-C}$ are functions of $\theta$?.
     It's actually not clear why pareto optimality is needed in Lemma 2 at all because all that really is being leveraged is that fact that the for a gradient inner product replacing one of the terms that was an average with a min lower bounds the product. It doesn't really say anything about finding a solution where improving on one domain must worsen performance on another (from the pareto optimality definition).

  4. Results
     These do appear to indicate an improvement over prior methods, however this is also partly affected by the issues from above with Fig (5) again showing a single run, admittedly with better correlation that that described in Fig (3), but with the training dynamics confusing things without sufficient explanation/discussion.
     For the efficiency demonstration, why is this shown only for VLCS? Are the gains not realised for other data?
     It would have been useful to see average gains across all the data explored in this work to see if the findings were consistent.

Overall, it appears that the idea and work has merit, but the exposition and discussion of what has been done needs a bit more work.

**Questions:**

The main ones are raised in the weakness section above to do with explaining the issues with prior work and the formulation of POGM itself.

Beyond this, some minor other questions/comments
1. Why different notations for source domains and training datasets (D_i vs S_i?)
2. Figure/plot axes labels and other text needs to be more legible and clear

---

### Meta-Review · Area_Chair_kCLd · 2024-12-20

**Metareview:**

This paper proposed a domain generalization method, which is built upon pareto optimality gradient matching. The authors show that it addressed the issues of gradient fluctuations and gradient magnitude elimination. The reviewers raised many concerns, including missing baselines and poor writing. The authors didn't submit a rebuttal so all reviewers keep the negative score in the end. I recommended a rejection.

**Additional Comments On Reviewer Discussion:**

No rebuttal was given so no further discussion.

---

### Decision · Program_Chairs · 2025-01-22

Reject